# On the Zeros of the Big $q$-Bessel Functions and Applications

**Fethi Bouzeffour** [1,*] 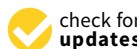**, Hanene Ben Mansour** [2] **and Mubariz Garayev** [3]

[1] Department of Mathematics, College of Sciences, King Saudi University, P. O Box 2455, Riyadh 11451, Saudi Arabia

[2] Department of Mathematics, Faculty of Sciences of Bizerte, University of Carthage, Zarzouna 7021, Tunisia; benmansourhanen52@yahoo.com

[3] Department of Mathematics, Faculty of Science, King Saud University, Riyadh 11451, Saudi Arabia; mgarayev@ksu.edu.sa

\* Correspondence: fbouzaffour@ksu.edu.sa

**Abstract:** This paper deals with the study of the zeros of the big $q$-Bessel functions. In particular, we prove new orthogonality relations for functions which are similar to the one for the classical Bessel functions. Also we give some applications related to the sampling theory.

**Keywords:** basic hypergeometric functions; completeness of sets of functions; interpolation

## 1. Introduction

The classical Bessel functions $J_\alpha(x)$ which are defined by [1]

$$J_\alpha(x) := \frac{(x/2)^\alpha}{\Gamma(\alpha+1)} \, {}_0F_1\left(\begin{array}{c} - \\ \alpha+1 \end{array} ; -\frac{x^2}{4}\right),$$

satisfy the orthogonality relations

$$\int_0^1 J_\alpha(j_{\alpha n}x)J_\alpha(j_{\alpha m}x)xdx = \frac{\delta_{nm}}{2}J_{\alpha+1}^2(j_{\alpha n}x), \tag{1}$$

where $\{j_{\alpha n}\}_{n\in\mathbb{N}}$ are the zeros of $J_\alpha(x)$.

Moreover, a function $f \in L^2((0,1); xdx)$, can be represented as the Fourier-Bessel series

$$f(x) = \sum_{n=0}^\infty c_n J_\alpha(j_{\alpha n}x), \tag{2}$$

where

$$c_n = \frac{2}{J_{\alpha+1}^2(j_{\alpha n}x)} \int_0^1 f(x)J_\alpha(j_{\alpha n}x)xdx. \tag{3}$$

In the literature there are many basic extensions of the Bessel functions $J_\alpha(x)$. The oldest one was introduced by Jackson in 1903–1905 and rewritten in modern notation by Ismail [2]. Other $q$-analogues can be obtained as formal limits of the three $q$-analogues of Jacobi polynomials; i.e., of little $q$-Jacobi polynomials, big $q$-Jacobi polynomials and Askey-Wilson polynomials. For this reason we propose to speak about little $q$-Bessel functions, big $q$-Bessel functions and AW type $q$-Bessel functions for the corresponding limit cases.

Recently, Koelink and Swarttouwn established orthogonality relations for the little $q$-Bessel (see [3,4]). Other orthogonality relations for Askey-Wilson functions were founded by Bustoz and Suslov (see, [5]). In this paper we discuss a new orthogonality relations for the big $q$-Bessel functions [6]

$$J_\alpha(x, \lambda; q^2) = {}_1\phi_1 \left( \begin{array}{c} -1/x^2 \\ q^{2\alpha+2} \end{array} \middle| q^2; \lambda^2 x^2 q^{2\alpha+2} \right). \tag{4}$$

In this work we show that all zeros of the big $q$-Bessel function $J_\alpha(x, \lambda; q^2)$ are real and simple. Further, using a similar technique as Bergweiler-Hayman [7] we derive an explicit asymptotic formula for these zeros, which is denoted $j_n^\alpha$:

$$j_n^\alpha = q^{-n-\alpha} a^{-1} (1 + O(q^n)).$$

In signal processing it is known that the space of band-limited signals is characterized as the set of all functions of $L^2(\mathbb{R})$ whose Fourier transforms have supports contained in $[-\pi, \pi]$, see [8,9]. The classical sampling theorem of Whittaker-Kotelnikov-Shannon (WKS), states that band-limited functions can be recovered from their values at the integers. In this work we provide a $q$-version of the sampling theorem of Whittaker-Kotelnikov-Shannon, and $q$-type band-limited signals which are defined in terms of Jackson's $q$-integral. The sampling points are the zeros of $J_\alpha(x, \lambda; q^2)$.

The paper is organized as follows: in Section 2, we define the big $q$-Bessel function, we give some recurrence relations and we prove that the big $q$-Bessel function is an eigenfunction of a $q$-difference equation of second order. Section 3, is devoted to study of the zeros of the big $q$-Bessel functions. In Section 4, we show that the set of functions $\{J_{\alpha+1}(a, j_n^\alpha; q^2)\}_{n=1}^\infty$ is a complete orthogonal system in $L_q^2((0, 1); w_\alpha(x; q) d_q t)$. Finally, in the last section, we give a $q$-version of the sampling theorem in the points $j_n^\alpha$.

## 2. The Big $q$-Bessel Functions

For the convenience of the reader, we provide a summary of the notations and definitions used in this paper.

Let $0 < q < 1$, $a \in \mathbb{C}$ and $n \in \mathbb{N}$; the $q$-shifted factorials are defined by [10]:

$$(a; q)_0 := 1, \quad (a; q)_n := \prod_{i=0}^{n-1} (1 - aq^i), \quad (a_1, \dots, a_k; q)_n := \prod_{j=1}^k (a_j; q)_n. \tag{5}$$

We also denote

$$(a; q)_\infty := \Pi_{n=0}^\infty (1 - aq^n).$$

The basic hypergeometric series ${}_r\phi_r$ is defined by [10]

$${}_r\phi_r \left( \begin{array}{c} a_1, \dots, a_r \\ b_1, \dots, b_r \end{array} \middle| q, z \right) := \sum_{k=0}^{+\infty} \frac{(a_1, \dots, a_r; q)_k}{(q, b_1, \dots, b_r; q)_k} (-1)^k q^{\binom{k}{2}} z^k. \tag{6}$$

The $q$-integral of a continuous function $f$ on $[0, a]$ $(a > 0)$ is defined by

$$\int_0^a f(t) d_q t = (1 - q) a \sum_{n=0}^\infty f(aq^n) q^n. \tag{7}$$

We introduce $q$-integration by parts. This will involve backward and forward $q$-derivatives

$$(D_q f)(x) = \frac{f(x) - f(qx)}{(1-q)x}, \quad (D_{1/q} f)(x) = q \frac{f(x/q) - f(x)}{(1-q)x}, \quad x \neq 0. \tag{8}$$

and $(D_q f)(0) = (D_{1/q} f)(0) = f'(0)$ provided $f'(0)$ exists.

If $f$ and $g$ are continuous on $[0, a]$　$(a \geq 0)$ then

$$\int_a^b D_{1/q} g(x) f(x) d_q x = q[f(x) g(q^{-1}x)]_a^b - q \int_a^b D_q f(x) g(x) d_q x. \tag{9}$$

The big $q$-Bessel functions are defined by [6]

$$
\begin{aligned}
J_\alpha(x, \lambda; q^2) &= {}_1\phi_1\left( \left. \begin{matrix} -1/x^2 \\ q^{2\alpha+2} \end{matrix} \right| q^2; \lambda^2 x^2 q^{2\alpha+2} \right), \\
&= \sum_{k=0}^\infty \frac{(-1)^k q^{2\binom{k}{2}+2k(\alpha+1)}}{(q^2, q^{2\alpha+2}; q^2)_k} \lambda^{2k} \Pi_{j=0}^{k-1}(x^2 + q^{2j}).
\end{aligned} \tag{10}
$$

For $\alpha > -1$, the functions $J_\alpha(x, \lambda; q^2)$ are analytic in $\mathbb{C}$ in their variables $x$ and $\lambda$ and satisfy

$$\lim_{q \to 1} J_\alpha\left( \frac{x}{1-q^2}, (1-q^2)^2 \lambda; q^2 \right) = \frac{\Gamma(\alpha+1)}{(\lambda x)^\alpha} J_\alpha(2\lambda x). \tag{11}$$

**Proposition 1.** *The big q-Bessel functions satisfy the following recurrence difference relations*

$$D_q J_\alpha(x, \lambda; q^2) = -\frac{\lambda^2 q^{2\alpha+2} x}{(1-q)(1-q^{2\alpha+2})} J_{\alpha+1}(x, \lambda; q^2), \tag{12}$$

$$D_{1/q}\left( w_{\alpha+1}(x; q) J_{\alpha+1}(x, \lambda; q^2) \right) = qx \frac{1-q^{2\alpha+2}}{1-q} w_\alpha(x; q) J_\alpha(x, \lambda; q^2), \tag{13}$$

*where*

$$w_\alpha(x; q) = \frac{(-x^2 q^2; q^2)_\infty}{(-x^2 q^{2\alpha+2}; q^2)_\infty}. \tag{14}$$

**Proof.** A simple computation shows that

$$D_q\left( (-1/x^2; q^2)_k x^{2k} \right) = \frac{1-q^{2k}}{1-q} (-1/x^2; q^2)_{k-1} x^{2k-1}.$$

Hence,

$$
\begin{aligned}
D_q J_\alpha(x, \lambda; q^2) &= \sum_{k=1}^\infty \frac{(-1)^k q^{2\binom{k}{2}+2k(\alpha+1)}}{(q^2, q^{2\alpha+2}; q^2)_k} \lambda^{2k} D_q\left( (-1/x^2; q^2)_k x^{2k} \right), \\
&= \frac{1}{(1-q)(1-q^{2\alpha+1})} \sum_{k=1}^\infty \frac{(-1)^k q^{2\binom{k}{2}+2k(\alpha+1)}}{(q^2, q^{2\alpha+4}; q^2)_{k-1}} \lambda^{2k} (-1/x^2; q^2)_{k-1} x^{2k-1}.
\end{aligned} \tag{15}
$$

Then, we obtain after making the change $k \to k+1$ in the second member of (15).

$$
\begin{aligned}
D_q J_\alpha(x, \lambda; q^2) &= -\frac{\lambda^2 q^{2\alpha+2} x}{(1-q)(1-q^{2\alpha+2})} \sum_{k=0}^\infty \frac{(-1)^k q^{2\binom{k}{2}+2k(\alpha+2)}}{(q^2, q^{2\alpha+4}; q^2)_k} (-1/x^2; q^2)_k (\lambda x)^{2k}, \\
&= -\frac{\lambda^2 q^{2\alpha+2} x}{(1-q)(1-q^{2\alpha+2})} J_{\alpha+1}(x, \lambda; q^2).
\end{aligned}
$$

On the other hand, from the following relation

$$D_{1/q}\left( w_{\alpha+1}(x; q)(-1/x^2; q^2)_k x^{2k} \right) = \frac{q^{-2k}(q^{2\alpha+2+2k} - 1)}{1-q^{-1}} w_\alpha(x; q)(-1/x^2; q^2)_k x^{2k+1}.$$

Then we get

$$
\begin{aligned}
&D_{1/q}\left(w_{\alpha+1}(x;q)J_{\alpha+1}(x,\lambda;q^2)\right)\\
&=\quad -\frac{x(1-q^{2\alpha+2})}{(1-q^{-1})}w_\alpha(x;q)\sum_{k=0}^{\infty}\frac{(-1)^k q^{2\binom{k}{2}+2k(\alpha+1)}}{(q^2,q^{2\alpha+2};q^2)_k}(-1/x^2;q^2)_k(\lambda x)^{2k},\\
&=\quad -\frac{x(1-q^{2\alpha+2})}{1-q^{-1}}w_\alpha(x;q)J_\alpha(x,\lambda;q^2).
\end{aligned}
$$

□

The trigonometric functions $\sin(x)$ and $\cos(x)$ are related to the Bessel function $J_\alpha(x)$ by

$$
\cos(x)=\sqrt{\frac{\pi x}{2}}J_{-\frac{1}{2}}(x),
$$

$$
\sin(x)=\sqrt{\frac{\pi x}{2}}J_{\frac{1}{2}}(x).
$$

Similarly, there are two $q$-trigonometric functions associated to the big $q$-Bessel function given

$$
\begin{aligned}
\cos(x,\lambda;q^2)&=\quad J_{-1/2}(x,\lambda;q^2),\\
&=\quad \sum_{k=0}^{\infty}\frac{(-1)^k q^{2\binom{k}{2}+k}}{(q;q)_{2k}}(-1/x^2;q^2)_k(\lambda x)^{2k},
\end{aligned}
$$

and

$$
\begin{aligned}
\sin(x,\lambda;q^2)&=\quad \frac{1}{1-q}J_{1/2}(x,\lambda;q^2),\\
&=\quad \sum_{k=0}^{\infty}\frac{(-1)^k q^{2\binom{k}{2}+3k}}{(q;q)_{2k+1}}(-1/x^2;q^2)_k(\lambda x)^{2k}.
\end{aligned}
$$

In particular, we have

$$
D_q\cos(x,\lambda;q^2)=-\frac{\lambda^2 qx}{1-q}\sin(x,\lambda;q^2).
$$

**Theorem 1.** *The big q-Bessel function is solution of the q-difference equation:*

$$
\frac{1}{xw_\alpha(x;q)}D_{q^{-1}}\left(\frac{w_{\alpha+1}(x;q)}{x}D_qJ_\alpha(x,\lambda;q^2)\right)=\frac{-\lambda^2 q^{2\alpha+3}}{(1-q)^2}J_\alpha(x,\lambda;q^2).
$$

*where $w_\alpha(x;q)$ is defined in* (14).

**Proof.** By (12) we have

$$
J_{\alpha+1}(x,\lambda;q^2)=-\frac{(1-q)(1-q^{2\alpha+2})}{\lambda^2 q^{2\alpha+2}x}D_qJ_\alpha(x,\lambda;q^2).
$$

Then the result is achieved by (13). □

**Proposition 2.** *The big q-Bessel functions satisfy the recurrence relations*

*(i)*

$$J_{\alpha+1}(x,\lambda;q^2) = \frac{(1-q^{2\alpha+2})}{\lambda^2 q^{2\alpha}(q^{2\alpha+2}x^2+1)}\big[(1-q^{2\alpha}-\lambda^2 q^{2\alpha}x^2)J_\alpha(x,\lambda;q^2)$$
$$-(1-q^{2\alpha})J_{\alpha-1}(x,\lambda;q^2)\big].$$

*(ii)*

$$J_{\alpha+1}(xq^{-1},\lambda;q^2) = \frac{(1-q^{2\alpha+2})}{\lambda^2 q^{2\alpha}(1+x^2)}\big[(1-q^{2\alpha})J_\alpha(x,\lambda;q^2)-J_{\alpha-1}(x,\lambda;q^2)\big].$$

**Proof.**

(i)   Accordingly to (12) and (13), we get

$$-x^2 q^2(1-q^{2\alpha})J_{\alpha-1}(xq,\lambda;q^2) = (1+x^2 q^{2\alpha+2})J_\alpha(xq,\lambda;q^2)$$
$$-(1+x^2 q^2)J_\alpha(x,\lambda;q^2), \tag{16}$$

and

$$J_\alpha(xq,\lambda;q^2) = J_\alpha(x,\lambda;q^2) + \frac{\lambda^2 x^2 q^{2\alpha+2}}{1-q^{2\alpha+2}}J_{\alpha+1}(x,\lambda;q^2). \tag{17}$$

Combining (16) and (17), we get

$$J_{\alpha+1}(x,\lambda;q^2) = \frac{(1-q^{2\alpha+2})}{\lambda^2 q^{2\alpha}(q^{2\alpha+2}x^2+1)}\big[(1-q^{2\alpha}-\lambda^2 q^{2\alpha}x^2)J_\alpha(x,\lambda;q^2)-(1-q^{2\alpha})J_{\alpha-1}(x,\lambda;q^2)\big].$$

(ii)   Similarly, the equality (12) gives us

$$J_\alpha(x,\lambda;q^2) = J_\alpha(xq,\lambda;q^2) - \frac{\lambda^2 x^2 q^{2\alpha+2}}{1-q^{2\alpha+2}}J_{\alpha+1}(x,\lambda;q^2). \tag{18}$$

By virtue of (16) and (18), we obtain

$$J_{\alpha+1}(xq^{-1},\lambda;q^2) = \frac{(1-q^{2\alpha+2})}{\lambda^2 q^{2\alpha}(1+x^2)}\big[(1-q^{2\alpha})J_\alpha(x,\lambda;q^2)-J_{\alpha-1}(x,\lambda;q^2)\big].$$

□

## 3. On the Zeros of the Big $q$-Bessel Functions

By using a similar method, as in [3], we prove in this section that the big $q$-Bessel function has infinite simple zeros on the real line, and by an explicit evaluation of a $q$-integral, we establish new orthogonality relations for this function.

**Proposition 3.** *Let $\alpha > -1$ and $a > 0$. For every $\lambda, \mu \in \mathbb{C} \setminus \{0\}$, we have*

$$(\lambda^2-\mu^2)\int_0^a J_{\alpha+1}(x,\lambda;q^2)J_{\alpha+1}(x,\mu;q^2)xw_\alpha(x;q)d_q x \tag{19}$$

$$= \frac{(1-q)(1-q^{2\alpha+2})}{q^{2\alpha+2}}w_\alpha(a;q)\big[J_{\alpha+1}(aq^{-1},\mu;q^2)J_\alpha(a,\lambda;q^2)-J_{\alpha+1}(aq^{-1},\lambda;q^2)J_\alpha(a,\mu;q^2)\big].$$

**Proof.** Using the *q*-integration by parts Equation (9) and Relations (12) and (13), we get

$$\int_0^a J_\alpha(x, \lambda; q^2) J_\alpha(x, \mu; q^2) x w_\alpha(x; q) d_q x = \frac{1-q}{1-q^{2\alpha+2}} \left[ x w_\alpha(x; q) J_{\alpha+1}(xq^{-1}, \lambda; q^2) J_\alpha(x, \mu; q^2) \right]_0^a$$

$$+ \frac{\lambda^2 q^{2\alpha+2}}{(1-q^{2\alpha+2})^2} \int_0^a J_{\alpha+1}(x, \lambda; q^2) J_{\alpha+1}(x, \mu; q^2) x w_{\alpha+1}(x; q) d_q x.$$

Then, the interchanging of $\lambda$ and $\mu$ in the last equation yields a set of two equations, which can be solved easily. $\square$

**Corollary 1.** *Let $\alpha > -1$ and $a > 0$. The zeros of the function $J_\alpha(a, \lambda; q^2)$ are real.*

**Proof.** Suppose $\lambda \neq 0$ is a zero of $\lambda \to J_\alpha(a, \lambda; q^2)$. We have

$$J_\alpha(a, \overline{\lambda}; q^2) = \overline{J_\alpha(a, \lambda; q^2)} = 0.$$

For $\mu = \overline{\lambda}$ Equation (3) yields

$$(\lambda^2 - \overline{\lambda}^2) \int_0^a |J_{\alpha+1}(x, \lambda; q^2)|^2 x w_{\alpha+1}(x; q) d_q x = 0.$$

Now $\lambda^2 = \overline{\lambda}^2$ if and only if $\lambda \in \mathbb{R}$ or $\lambda \in i\mathbb{R}$, then in all other cases we have

$$\int_0^a |J_{\alpha+1}(x, \lambda; q^2)|^2 x w_{\alpha+1}(x; q) d_q x = 0. \tag{20}$$

Using the definition of the *q*-integral we get

$$\int_0^a x |J_{\alpha+1}(x, \lambda; q^2|^2 d_q x w_{\alpha+1}(x; q) x$$

$$= (1-q) a^2 \sum_{k=0}^\infty \frac{q^{2k} (-a^2 q^{2k+2}; q^2)_\infty}{(-a^2 q^{2k+2\alpha+4}; q^2)_\infty} |J_{\alpha+1}(aq^k, \lambda; q^2)|^2;$$

then,

$$J_{\alpha+1}(aq^k, \lambda; q^2) = 0, \ k \in \mathbb{N},$$

and $J_{\alpha+1}(., \lambda; q^2)$ defines an analytic function on $\mathbb{C}$. Hence, $J_{\alpha+1}(., \lambda; q^2) = 0$. Now if $\lambda = i\mu$, with $\mu \in \mathbb{R}$, then we have

$$J_\alpha(a, i\mu; q^2) = \sum_{k=0}^\infty \frac{q^{2\binom{k}{2}+2k(\alpha+1)}}{(q^2, q^{2\alpha+2}; q^2)_k} (-\frac{1}{a^2}; q^2)_k (a\mu)^{2k}.$$

For $\alpha > -1$ this expression cannot be zero, which proves the corollary. $\square$

To obtain an expression for the *q*-integral in Equation (3) with $\lambda = \mu$, we use l'Hopital's rule. The result is

$$\int_0^a \left( J_{\alpha+1}(x, \lambda; q^2) \right)^2 x w_{\alpha+1}(x; q) d_q x = \frac{(q-1)(1-q^{2\alpha+2})(-a^2; q^2)_\infty}{2\lambda q^{2\alpha+2}(-a^2 q^{2\alpha+2}; q^2)_\infty}$$

$$\left[ J_\alpha(a, \lambda; q^2) \left[ \frac{\partial J_{\alpha+1}}{\partial \mu}(aq^{-1}, \mu; q^2) \right]_{\mu=\lambda} - J_{\alpha+1}(aq^{-1}, \lambda; q^2) \left[ \frac{\partial J_\alpha}{\partial \mu}(a, \mu; q^2) \right]_{\mu=\lambda} \right].$$

This formula is reduced to

$$\int_0^a x (J_{\alpha+1}(x, \lambda; q^2))^2 x w_{\alpha+1}(x; q) d_q x \tag{21}$$

$$= \frac{(1-q)(1-q^{2\alpha+2})}{2\lambda q^{2\alpha+2}} \frac{(-a^2; q^2)_\infty}{(-a^2 q^{2\alpha+2}; q^2)_\infty} J_{\alpha+1}(aq^{-1}, \lambda; q^2) \left[ \frac{\partial J_\alpha}{\partial \mu}(a, \mu; q^2) \right]_{\mu=\lambda},$$

for $\lambda \neq 0$, a real zero of $J_\alpha(a, \lambda; q^2)$.

**Lemma 1.** *The non-zero real zeros of $\lambda \to J_\alpha(a, \lambda; q^2)$, with $\alpha > -1$, are simple zeros.*

**Proof.** Let $\lambda$ be a non-zero real zero of $J_\alpha(a, \lambda; q^2)$, with $\alpha > -1$. The integral

$$\int_0^a |J_{\alpha+1}(x, \lambda; q^2)|^2 x w_{\alpha+1}(x; q) d_q x = \int_0^a (J_{\alpha+1}(x, \lambda; q^2))^2 x w_{\alpha+1}(x; q) d_q x$$

is strictly positive. If it were zero, this would imply that the big $q$-Bessel function is identically zero as in the proof of Corollary 1. Hence, (21) implies that

$$\left[ \frac{\partial J_\alpha}{\partial \mu}(a, \mu; q^2) \right]_{\mu=\lambda} \neq 0,$$

which proves the lemma. □

Recall that the order $\rho(f)$ of an entire function $f(z)$, see [11,12], is given by

$$\rho(f) = \limsup_{r \to \infty} \frac{\ln \ln M(r;\, f)}{\ln r}, \qquad M(r;\, f) = \max_{|z| \le r} |f(z)|.$$

**Lemma 2.** *For $\alpha > -\frac{1}{2}$ and $a \in \mathbb{R}$, the big $q$-Bessel function $\lambda \to J_\alpha(a, \lambda; q^2)$ has infinitely many zeros.*

**Proof.** We have

$$J_\alpha(a, \lambda; q^2) = \sum_{n=0}^{\infty} a_n q^{n^2} (\lambda a)^{2n},$$

with

$$a_n = \frac{(-1)^n q^{(2\alpha+1)n}}{(q^2, q^{2\alpha+2}; q^2)_n} \left( -\frac{1}{a^2}; q^2 \right)_n.$$

By [12], Theorem 1.2.5, it suffices to show that $\rho(J_\alpha(a, \lambda; q^2)) = 0$.
Since $\alpha > -1/2$, we have

$$\lim_{n \to \infty} a_n = 0,$$

there exists $C > 0$ such that $|a_n| < C$, and for $|\lambda| < r$, we have

$$\begin{aligned} M(r, \lambda \to J_\alpha(a, \lambda; q^2)) &\le C \sum_{n=0}^{\infty} q^{n^2}(ar)^{2n}, \\ &< C \sum_{n=-\infty}^{+\infty} q^{n^2}(ar)^{2n}. \end{aligned}$$

The Jacobi 's triple identity (see, [10]) leads to

$$M(r, \lambda \to J_\alpha(a, \lambda; q^2)) < C(q^2, -(ar)^2 q, -q/(ar)^2; q^2)_\infty.$$

Set $r = \frac{q^{-(N+\varepsilon)}}{a}$, for $-\frac{1}{2} \le \varepsilon < \frac{1}{2}$ and $N = 0, 1, 2, \dots$ . Clearly

$$\begin{aligned} (-(ar)^2 q; q^2)_\infty &= (-q^{-2(N+\varepsilon)+1}; q^2)_\infty, \\ &= (-q^{1-2\varepsilon} q^{-2N}; q^2)_\infty. \end{aligned}$$

We have

$$(-(ar)^2 q; q^2)_\infty = (-q^{1-2\varepsilon}; q^2)_\infty (-q^{1-2\varepsilon-2N}; q^2)_N$$

and

$$
\begin{aligned}
(-(ar)^2 q; q^2)_\infty &= q^{-(N^2+2N\varepsilon)}(-q^{2\varepsilon+1}; q^2)_N(-q^{1-2\varepsilon}; q^2)_\infty, \\
&\leq q^{-(N^2+2N\varepsilon)}(-q^{2\varepsilon+1}; q^2)_\infty(-q^{1-2\varepsilon}; q^2)_\infty;
\end{aligned}
$$

additionally,

$$
\begin{aligned}
(-q/(ar)^2; q^2)_\infty &= (-q^{2(N+\varepsilon)+1}; q^2)_\infty, \\
&= (-1; q^2)_\infty.
\end{aligned}
$$

Hence

$$
\frac{\ln \ln M(r, \lambda \to J_\alpha(a, \lambda; q^2))}{\ln q^{-(N+\varepsilon)}},
$$

$$
\leq \frac{\ln \ln A q^{-N(N+2\varepsilon)}}{\ln q^{-(N+\varepsilon)}},
$$

$$
= \frac{\ln N(N+2\varepsilon) + \ln\left(\frac{\ln A}{N(N+2\varepsilon)} - \ln q\right)}{-(N+\varepsilon)\ln q},
$$

where

$$
A = C(-q^{1-2\varepsilon}, -q^{1+2\varepsilon}, -1; q^2)_\infty.
$$

This implies

$$
\rho(\lambda \to J_\alpha(a, \lambda; q^2)) = 0.
$$

□

For $\alpha > -\frac{1}{2}$, we order the positive zeros of $\lambda \to J_\alpha(a, \lambda; q^2)$ as

$$
0 < j_1^\alpha < j_2^\alpha < j_3^\alpha < \dots.
$$

Next, we derive an explicit asymptotic formula for the zeros of the big $q$-Bessel function $\lambda \to J_\alpha(a, \lambda; q^2)$, using the same technique of Bergweiler-Hayman [7] and Annaby-Mansour [13,14]. We start with two preliminary results.

**Theorem 2.** *If $\alpha > -3/2$ and $j_n^\alpha$ are the positive zeros of $J_\alpha(a, .; q^2)$ then we have for the sufficiently large $n$,*

$$
j_n^\alpha = q^{-n-\alpha} a^{-1}(1 + O(q^n)), \quad |a^{-1}| < 1. \tag{22}
$$

**Proof.** The proof is similar to the proof of Theorem 2.1 in [14] and is omitted. □

Additionally, we can prove the following

**Theorem 3.** *If $\alpha > -3/2$ and $x_{n,\alpha}$ are the positive zeros of $J_\alpha(., \lambda; q^2)$, then we have for sufficiently large $n$,*

$$
x_{n,\alpha} = q^{-n-\alpha} \lambda^{-1}(1 + O(q^n)), \quad \lambda \in \mathbb{C}\backslash\{0\}. \tag{23}
$$

In order to investigate the asymptotic of the functions $J_\alpha(a, \lambda; q^2)$, we define the suitable sets of annuli $\{B_{n,\alpha}\}$ in terms of the zeros $\lambda_{n,\alpha}$ such that for every $n \in \mathbb{Z}^+$, $B_{n,\alpha}$ intersects with $B_{n+1,\alpha}$ only on the boundary. The sets of annuli are constructed such that the annulus $B_{n,\alpha}$ contains only the zeros $\pm j_n^\alpha$ of $J_\alpha(a, \lambda; q^2)$. Then we study the behavior of $J_\alpha(a, \lambda; q^2)$ in $B_{n,\alpha}$ when $n$ is large enough. Let

$$
\beta_{n,\alpha} := \log\left(j_n^\alpha/j_{n+1}^\alpha\right) / \log q^2, \quad n \in \mathbb{Z}^+. \tag{24}
$$

Then $\beta_{n,\alpha}$ is a decreasing nonnegative sequence. Moreover, from (22) we have

$$\beta_{n,\alpha} = 1 + \frac{\log\left(\dfrac{1+O(q^{2n})}{1+O(q^{2n+2})}\right)}{\log q^2} = 1 + \frac{\log\left(1+O(q^{2n})\right)}{\log q^2},$$

for sufficiently large $n$. That is, $\lim_{n\to\infty} \beta_{n,\alpha} = 1$. Therefore

$$0 < \beta_\alpha := \inf_{n\in\mathbb{Z}^+} \beta_{n,\alpha} = 1. \tag{25}$$

We define the positive sequences $\{c_{n,\alpha}\}_{n=1}^\infty$ and $\{d_{n,\alpha}\}_{n=1}^\infty$ :

$$c_{m,\alpha} := \begin{cases} \beta_{n,\alpha} + 1, & \text{if } \beta_{n,\alpha} \neq 1, \\[2mm] \beta_\alpha, & \text{if } \beta_{n,\alpha} = 1, \end{cases} \tag{26}$$

and

$$d_{1,\alpha} := 1, \quad d_{n+1,\alpha} := \begin{cases} \beta_{n,\alpha} - 1, & \text{if } \beta_{n,\alpha} \neq 1, \\[2mm] 1, & \text{if } \beta_{n,\alpha} = 1, \end{cases} \tag{27}$$

where $n \geqslant 1$. We can easily verify that

$$j_n^\alpha q^{-c_{n,\alpha}} = j_{n+1}^\alpha q^{d_{n+1,\alpha}}, \quad n \geqslant 1. \tag{28}$$

Put

$$B_{n,\alpha} := \left\{z \in \mathbb{C} : \ j_n^\alpha q^{d_{n,\alpha}} \leqslant |\lambda| \leqslant j_n^\alpha q^{-c_{n,\alpha}}\right\}, \quad n \geqslant 1, \tag{29}$$

dividing the region $\{z \in \mathbb{C} : \ |\lambda| \geqslant q j_1^\alpha\}$ into annuli with common boundaries. Now we introduce the asymptotic of $J_\alpha(a, \lambda; q^2)$ in the set of annuli $\{B_{n,\alpha}\}_{n=1}^\infty$, when $m$ is large enough.

**Theorem 4.** *Assume that $|\lambda| \geqslant q j_1^\alpha$ and $B_{n,\alpha}$, $n \geqslant 1$, is the annulus defined in (29). Then we have the asymptotic relation*

$$\log|J_\alpha(a, \lambda; q)| = -2\frac{\log\left|\lambda\, q^{2\alpha} a^2\right|^2}{\log q} - 2\log|\lambda\, a^2 q^{2\alpha}| + \log\left|1 - \frac{\lambda^2}{(j_n^\alpha)^2}\right| + O(1), \ \lambda \in B_{n,\alpha}^{(3)}, \tag{30}$$

*uniformly when $m$ is sufficiently large.*

**Proof.** The proof is similar to the proof of Theorem 3.2 in [15] and is omitted. $\square$

**Theorem 5.** *Let $\lambda \in \mathbb{C}$ be fixed, and let $|arg(-\lambda x)| < \pi$ and $|\lambda x| \to \infty$. The asymptotic*

$$J_\alpha(x, \lambda; q^2) = \frac{(\lambda^2 x^2 q^{2\alpha+2}; q^2)_\infty}{(q^{2\alpha+2}; q^2)_\infty}\left\{1 + \sum_{n=1}^\infty b_n(\lambda x)^{-2n}\right\}, \tag{31}$$

*holds true with*

$$b_n = q^{2n} \sum_{k=1}^n \frac{q^{k(2-n+k)}}{(q^2; q^2)_k}(-\lambda^2; q^2)_k \begin{bmatrix} n-1 \\ k-1 \end{bmatrix}_{q^2}. \tag{32}$$

**Proof.** Using the transformation, see [10],

$$(c; q)_\infty \, _1\phi_1(a; c; q, z) = (z; q)_\infty \, _1\phi_1(\frac{az}{c}; z; q, c), \tag{33}$$

we have

$$
J_\alpha(x,\lambda;q^2) = {}_1\phi_1\left(\frac{-1}{x^2};q^{2\alpha+2};q^2,\lambda^2 x^2 q^{2\alpha+2}\right),
$$

$$
= \frac{(\lambda^2 x^2 q^{2\alpha};q^2)_\infty}{(q^{2\alpha+2};q^2)_\infty} {}_1\phi_1(-\lambda^2;\lambda^2 x^2 q^{2\alpha+2},q^2,q^{2\alpha+2}),
$$

$$
= \frac{(\lambda^2 x^2 q^{2\alpha};q^2)_\infty}{(q^{2\alpha+2};q^2)_\infty} \sum_{k=0}^\infty \frac{(-\lambda^2;q^2)_k}{(q^2,q^{2\alpha+2}\lambda^2 x^2;q^2)_k} q^{k(k-1)}(-q^{2\alpha+2})^k.
$$

This representation shows that $J_\alpha(x,\lambda;q^2)$ is also an entire function in $x$ and $\lambda$. Using that

$$
(z;q)_k = (-1)^k z^k q^{k(k-1)/2}(1/z;1/q)_k,
$$

we obtain

$$
J_\alpha(x,\lambda;q^2) = \frac{(\lambda^2 x^2 q^{2\alpha};q^2)_\infty}{(q^{2\alpha+2};q^2)_\infty}\left\{1+\sum_{k=1}^\infty \frac{(-\lambda^2;q^2)_k}{(q^2;q^2)_k}\frac{(\lambda^2 x^2)^{-k}}{(q^{-2\alpha-2}/\lambda^2 x^2;1/q^2)_k}\right\}.
$$

Using also that

$$
\frac{1}{(1/z;1/q)_k} = \sum_{n=0}^\infty\sum_{n=0}^\infty z^{-n}q^{-n(k-1)}\begin{bmatrix}k+n-1\\n\end{bmatrix}_q,\quad |z|>q^{-k+1},
$$

we get

$$
J_\alpha(x,\lambda;q^2) = \frac{(\lambda^2 x^2 q^{2\alpha};q^2)_\infty}{(q^{2\alpha+2};q^2)_\infty}\left\{1+\sum_{n=1}^\infty(\lambda^2 x^2)^{+n}(q^2;q^2)_{n-1}q^{2n}\sum_{k=1}^n\frac{q^{(2\alpha+2)k}(-\lambda^2;q^2)_k q^{-2(k-1)(n-k)}}{(q^2;q^2)_k(q^2;q^2)_{n-k}(q^2;q^2)_{k-1}}\right\},
$$

$$
= \frac{(\lambda^2 x^2 q^{2\alpha};q^2)_\infty}{(q^{2\alpha+2};q^2)_\infty}\left\{1+\sum_{n=1}^\infty b_n(\lambda^2 x^2)^{-n}\right\},
$$

where

$$
b_n = q^{2n}\sum_{k=1}^n\frac{q^{k(2-n+k)}}{(q^2;q^2)_k}(-\lambda^2;q^2)_k\begin{bmatrix}n-1\\k-1\end{bmatrix}_{q^2}. \tag{34}
$$

□

## 4. Orthogonality Relation and Completeness

The Proposition 3 and Relation (3) are useful to state the orthogonality relations for the big $q$-Bessel functions.

**Proposition 4.** *Let $\alpha > -\frac{1}{2}$ and $0 < j_1^\alpha < j_2^\alpha < j_3^\alpha < \dots$ be the positive zeros of the big $q$-Bessel function $J_\alpha(1,\lambda;q^2)$. Then*

$$
\int_0^1 J_{\alpha+1}(x,j_n^\alpha;q^2)J_{\alpha+1}(x,j_m^\alpha;q^2)xw_{\alpha+1}(x;q)d_q x
$$

$$
= \frac{(1-q)(1-q^{2\alpha+2})}{2\lambda q^{2\alpha+2}}\frac{(-1;q^2)_\infty}{(-q^{2\alpha+2};q^2)_\infty}J_{\alpha+1}(q^{-1},j_n^\alpha;q^2)\left[\frac{\partial J_\alpha}{\partial \mu}(1,\mu;q^2)\right]_{\mu=j_n^\alpha}\delta_{n,m}. \tag{35}
$$

We consider the inner product giving by

$$
<f,g> = \int_0^1 f(t)\overline{g(t)}xw_{\alpha+1}(t;q)d_q t.
$$

Let

$$G(\lambda) = \int_0^1 g(x) J_{\alpha+1}(x, \lambda; q^2) x w_{\alpha+1}(x; q) d_q x$$

and

$$h(\lambda) = \frac{G(\lambda)}{J_\alpha(1, \lambda; q^2)}.$$

**Lemma 3.** *If $\alpha > -\frac{3}{2}$ and $g(x) \in L_q^2((0,1); x w_{\alpha+1}(x; q) d_q x)$, then $h(\lambda)$ is entire of order 0.*

**Proof.** We first show that $G(\lambda)$ is entire of order 0. From the definition of the $q$-integral, we have

$$G(\lambda) = (1 - q) \sum_{k=0}^{\infty} \frac{q^k (-q^{2k+2}; q^2)_\infty}{(-q^{2k+2\alpha+4}; q^2)_\infty} g(q^k) J_{\alpha+1}(q^k, \lambda; q^2) q^k. \tag{36}$$

The series (36) converges uniformly in any disk $|\lambda| \le R$. Hence $G(\lambda)$ is complete and we have

$$M(r; G) \le M(r; \lambda \to J_{\alpha+1}(x, \lambda; q^2)) \int_0^1 \frac{x(-x^2 q^2; q^2)_\infty}{(-x^2 q^{2\alpha+4}; q^2)_\infty} |g(x)| d_q x.$$

Since $\rho(\lambda \to J_{\alpha+1}(x, \lambda; q^2)) = 0$, we have that $\rho(G) = 0$.

Both the numerator and the denominator of $h(\lambda)$ are entire functions of order 0. If we write a factor of $G(\lambda)$ and $J_\alpha(1, \lambda; q^2)$ as canonical products, each factor of $J_\alpha(1, \lambda; q^2)$ that divides out with a factor of $G(\lambda)$, by hypothesis $h(\lambda)$ is thus entirety of order 0. $\square$

**Lemma 4.** *For $m = 0, 1, 2, \ldots$, the quotient $\frac{J_{\alpha+1}(q^m, \lambda; q^2)}{J_\alpha(1, \lambda; q^2)}$ is bounded on the imaginary axis.*

**Proof.** We will make use of the simple inequalities

$$
\begin{aligned}
(-q^{-2m}; q^2)_n q^{2nm} &= \prod_{j=0}^{n-1} (q^{2m} + q^{2j}) \\
&\le \prod_{j=0}^{\infty} (1 + q^{2j}) \\
&= (-1; q^2)_\infty, \quad m \in \mathbb{N},
\end{aligned}
$$

$$1 - q^{2n+2\alpha+2} > 1 - q^{2\alpha+2},$$

and

$$(-1; q^2)_n \ge 1.$$

We get for $\lambda = i\mu, \mu$ real,

$$
\begin{aligned}
J_{\alpha+1}(q^m, i\mu; q^2) &= \sum_{n=0}^{\infty} \frac{q^{2\binom{n}{2}+2n(\alpha+2)}}{(q^2, q^{2\alpha+4}; q^2)_n} (-q^{-2m}; q^2)_n q^{2nm} \mu^{2n}, \\
&\le \sum_{n=0}^{\infty} \frac{q^{2\binom{n}{2}+2n(\alpha+1)}}{(q^2, q^{2\alpha+2}; q^2)_n} \frac{1 - q^{2\alpha+2}}{1 - q^{2\alpha+2+2n}} (-1; q^2)_\infty \mu^{2n}, \\
&< \sum_{n=0}^{\infty} \frac{q^{2\binom{n}{2}+2n(\alpha+1)}}{(q^2, q^{2\alpha+2}; q^2)_n} (-1; q^2)_\infty \mu^{2n}.
\end{aligned}
$$

$$J_\alpha(1, i\mu; q^2) = \sum_{n=0}^{\infty} \frac{q^{2\binom{n}{2}+2n(\alpha+1)}}{(q^2, q^{2\alpha+2}; q^2)_n} (-1; q^2)_n \mu^{2n},$$

$$\geq \sum_{n=0}^{\infty} \frac{q^{2\binom{n}{2}+2n(\alpha+1)}}{(q^2, q^{2\alpha+2}; q^2)_n} \mu^{2n}.$$

Thus, we have

$$0 \leq \frac{J_{\alpha+1}(q^m, i\mu; q^2)}{J_\alpha(1, i\mu; q^2)} \leq (-1; q^2)_\infty.$$

□

**Theorem 6.** *For* $\alpha > -\frac{3}{2}$*, the system* $\{J_{\alpha+1}(x, j_n^\alpha; q^2)\}_{n=0}^{\infty}$ *is complete in* $L_q^2((0,1); xw_\alpha(x; q))$.

**Proof.** Lemma 3 implies that $h(i\mu)$ is bounded. Since $h(\lambda)$ is entire of order 0, we can apply one of the versions of the Phragmén-Lindelöf theorem, see [11] and Lemma 3 and conclude that $h(\lambda)$ is bounded in the entire $\lambda$-plane. Next, by Liouville's theorem we conclude that $h(\lambda)$ is constant. Say that $h(\lambda) = C$. We will prove that $C = 0$. Indeed, we have

$$G(\lambda) = C J_\alpha(1, \lambda; q^2),$$

$$
\begin{aligned}
G(\lambda) &= \int_0^1 \frac{a(-a^2 q^2; q^2)_\infty}{(-a^2 q^{2\alpha+4}; q^2)_\infty} g(a) J_{\alpha+1}(a, \lambda; q^2) d_q a, \\
&= (1-q) \sum_{k=0}^{\infty} \frac{q^{2k}(-q^{2k+2}; q^2)_\infty}{(-q^{2k+2\alpha+4}; q^2)_\infty} g(q^k) J_{\alpha+1}(q^k, \lambda; q^2), \\
&= (1-q) \sum_{k=0}^{\infty} \frac{q^{2k}(-q^{2k+2}; q^2)_\infty}{(-q^{2k+2\alpha+4}; q^2)_\infty} g(q^k) \sum_{n=0}^{\infty} \frac{(-1)^n q^{2\binom{n}{2}+2n(\alpha+2)}}{(q^2, q^{2\alpha+4}; q^2)_n} \left(-\frac{1}{q^{2k}}; q^2\right)_n q^{2kn} \lambda^{2n}, \\
&= (1-q) \sum_{n=0}^{\infty} \frac{(-1)^n q^{2\binom{n}{2}+2n(\alpha+2)}}{(q^2, q^{2\alpha+4}; q^2)_n} \sum_{k=0}^{\infty} \frac{q^{2k}(-q^{2k+2}; q^2)_\infty}{(-q^{2k+2\alpha+4}; q^2)_\infty} g(q^k) \left(-\frac{1}{q^{2k}}; q^2\right)_n q^{2kn} \lambda^{2n},
\end{aligned}
$$

and

$$J_\alpha(1, \lambda; q^2) = \sum_{n=0}^{\infty} \frac{(-1)^n q^{2\binom{n}{2}+2n(\alpha+1)}}{(q^2, q^{2\alpha+2}; q^2)_n} (-1; q^2)_n \lambda^{2n}.$$

It follows that

$$(1-q)\frac{(-1)^n q^{2\binom{n}{2}+2n(\alpha+1)}}{(q^2, q^{2\alpha+4}; q^2)_n} q^{2n} \sum_{k=0}^{\infty} \frac{q^{2k}(-q^{2k+2}; q^2)_\infty}{(-q^{2k+2\alpha+4}; q^2)_\infty} g(q^k) \left(-\frac{1}{q^{2k}}; q^2\right)_n q^{2kn}$$

$$= \frac{(-1)^n q^{2\binom{n}{2}+2n(\alpha+1)}}{(q^2, q^{2\alpha+2}; q^2)_n} (-1; q^2)_n C, \quad n = 0, 1, 2, \dots.$$

Dividing to common factors, we have

$$(1-q)\frac{1-q^{2\alpha+2}}{1-q^{2\alpha+2n+2}} q^{2n} \sum_{k=0}^{\infty} \frac{q^{2k}(-q^{2k+2}; q^2)_\infty}{(-q^{2k+2\alpha+4}; q^2)_\infty} g(q^k) \left(-\frac{1}{q^{2k}}; q^2\right)_n q^{2kn}$$

$$= (-1; q^2)_n C, \quad n = 0, 1, 2, \dots$$

and letting $n \to \infty$ gives

$$(-1; q^2)_\infty C = 0$$

hence,

$$C = 0.$$

We can now conclude that

$$G(\lambda) = 0,$$

or

$$\int_0^1 g(x) J_{\alpha+1}(x, j_n^\alpha; q^2) x w_{\alpha+1}(x; q) d_q x = 0.$$

We complete the proof with a simple argument that gives $g(q^m) = 0, m = 0, 1, \ldots$

If

$$G(\lambda) = 0,$$

then

$$\sum_{k=0}^\infty \frac{q^{2k}(-q^{2k+2}; q^2)_\infty}{(-q^{2k+2\alpha+2}; q^2)_\infty} g(q^k) (-\frac{1}{q^{2k}}; q^2)_n q^{2kn} = 0.$$

Letting $n \to \infty$ gives $g(1) = 0$. Then, dividing by $q^{2n}$ and again letting $n \to \infty$ gives $g(q) = 0$. Continuing this process, we have $g(q^m) = 0$, which completes the proof. □

Using the orthogonality Relation (35), we consider the big $q$-Fourier-Bessel series, $S_q^{(\alpha)}[f]$, associated with a function $f$,

$$S_q^{(\alpha)}[f](x) = \sum_{k=1}^\infty a_k(f) J_{\alpha+1}(x, j_k^\alpha; q^2), \tag{37}$$

with the coefficients $a_k$ given by

$$a_k(f) = \frac{1}{\mu_k} \int_0^1 f(t) J_{\alpha+1}(t, j_k^\alpha; q^2) t w_{\alpha+1}(t; q) d_q t, \tag{38}$$

where

$$\begin{aligned}
\mu_k &= \|J_{\alpha+1}(x, j_k^\alpha; q^2)\|_{L_q^2(0,1)}^2, \\
&= \int_0^1 [J_{\alpha+1}(x, j_k^\alpha; q^2)]^2 x w_{\alpha+1}(x; q) d_q x.
\end{aligned}$$

## 5. Sampling Theorem

The classical Kramer sampling is as follows [8,16]. Let $K(x; \lambda)$ be a function, continuous in $\lambda$ such that, as a function of $x$, $K(x; \lambda) \in L^2(I)$ for every real number $\lambda$, where $I$ is an interval of the real line. Assume that there exists a sequence of real numbers $\lambda_n$, with $n$ belonging to an indexing set contained in $\mathbb{Z}$, such that $K(x; \lambda_n)$ is a complete orthogonal sequence of functions of $L^2(I)$. Then for any $F$ of the form

$$F(\lambda) = \int_I f(x) K(x, \lambda) dx,$$

where $F \in L^2(I)$, we have

$$F(\lambda) = \lim_{N \to \infty} \sum_{|n| \le N} F(\lambda_n) S_n(\lambda), \tag{39}$$

with

$$S_n(t) = \frac{\int_I K(x, \lambda) \overline{K(x, \lambda_n)} dx}{\int_I |K(x, \lambda_n)|^2 dx}$$

The series (39) converges uniformly wherever $||K(.,\lambda)||_{L^2(I)}$ is bounded.

Now we give a *q*-sampling theorem for the *q*-integral transform of the form

$$F(\lambda) = \int_0^1 f(x)J_{\alpha+1}(x,\lambda;q^2)xw_{\alpha+1}(x;q)d_qx, \qquad f \in L_q^2(0,1), \qquad \alpha > -\frac{3}{2}.$$

**Theorem 7.** *Let* $f$ *be a function in* $L_q^2((0,1);xw_{\alpha+1}(x;q)d_qx)$. *Then the q-integral transform*

$$F(\lambda) = \int_0^1 f(x)J_{\alpha+1}(x,\lambda;q^2)xw_{\alpha+1}(x;q)d_qx, \qquad \alpha > -\frac{3}{2}. \tag{40}$$

*has the point-wise convergent sampling expansion*

$$F(\lambda) = \sum_{k=1}^{\infty} 2F(j_k^\alpha)\frac{j_k^\alpha J_{\alpha+1}(1,\lambda;q^2)}{(\lambda^2 - (j_k^\alpha)^2)[\frac{\partial J_{\alpha+1}}{\partial \lambda}(1,\lambda;q^2)]_{\lambda=j_k^\alpha}}. \tag{41}$$

*The series* (41) *converges uniformly over any compact subset of* $\mathbb{C}$.

**Proof.** Set $K(x,\lambda) = J_{\alpha+1}(x,\lambda;q^2)$, and $j_k^\alpha$ is the *k*-th positive zero of $J_{\alpha+1}(x,\lambda;q^2)$, and $\{J_{\alpha+1}(x,j_k^\alpha;q^2)\}_{k=1}^{\infty}$ is a complete orthogonal sequence of function in $L_q^2((0,1);xw_{\alpha+1}(x;q)d_qx)$. Then we get

$$F(\lambda) = \sum_{k=1}^{\infty} F(j_k^\alpha)\frac{\int_0^1 J_{\alpha+1}(x,j_k^\alpha;q^2)J_{\alpha+1}(x,\lambda;q^2)xw_{\alpha+1}(x;q)d_qx}{\int_0^1 xw_{\alpha+1}(x;q)|J_{\alpha+1}(x,j_k^\alpha;q^2)|^2xw_{\alpha+1}(x;q)d_qx}. \tag{42}$$

But $J_{\alpha+1}(x,\lambda;q^2)$ is analytic on $\mathbb{C}$, so is bounded on any compact subset of $\mathbb{C}$, and hence $||J_{\alpha+1}(x,\lambda;q^2)||_2$ is bounded. Substituting from (3) with $\mu = j_k^\alpha$ we obtained (42) and the theorem follows. □

**Example 1.** *Define a function f on* $[0,1]$:

$$f(t) = \begin{cases} \frac{1}{1-q} & t = 1, \\ 0 & otherwise. \end{cases}$$

*Then*

$$F(\lambda) = \int_0^1 f(x)J_{\alpha+1}(x,\lambda;q^2)xw_{\alpha+1}(x;q)d_qx = \frac{(-q^2;q^2)_\infty}{(-q^{2\alpha+4};q^2)_\infty}J_{\alpha+1}(1,\lambda;q^2).$$

*Thus, applying Theorem 7 gives*

$$\frac{1}{2} = \sum_{k=1}^{\infty} \frac{j_k^\alpha J_{\alpha+1}(1,j_k^\alpha;q^2)}{(\lambda^2 - (j_k^\alpha)^2)[\frac{\partial J_{\alpha+1}}{\partial \lambda}(1,\lambda;q^2)]_{\lambda=j_k^\alpha}}, \qquad \lambda \in \mathbb{C}\setminus\{\pm j_k^\alpha, k \in \mathbb{N}^*\}. \tag{43}$$

*We define the Paley-Wiener space related to the big q-Bessel function by*

$$PW_B = \{\mathcal{F}_B(f); \ f \in L_q^2((0,1);xw_{\alpha+1}(x;q)d_qx)\},$$

*where the finite big q-Hankel transform* $\mathcal{F}_B(f)$ *is defined by*

$$\mathcal{F}_B(f)(\lambda) = \int_0^1 f(x)J_{\alpha+1}(x,\lambda;q^2)xw_{\alpha+1}(x;q)d_qx, \qquad \alpha > -\frac{3}{2}. \tag{44}$$

*By quite similar arguments to those in the proof of* [17], *Theorem 1, we see that the space* $PW_B$ *equipped with the inner product*

$$< f,g >_{PW} = \int_0^1 (\mathcal{F}_Bf)(x)\overline{(\mathcal{F}_Bg)(x)}xw_{\alpha+1}(x;q)d_qx$$

*is a Hilbert space, and the finite big q-Hankel transform* (44) *becomes a Hilbert space isometry between* $L_q^2((0,1); xw_{\alpha+1}(x;q)d_qx)$ *and* $PW_A$. *Therefore, from (*[17]*, Theorem A) we deduce that the big q-Bessel function has an associated reproducing kernel.*

**Author Contributions:** Formal analysis, F.B., H.B.M. and M.G.; Methodology, F.B., H.B.M. and M.G.; Writing—original draft, F.B., H.B.M. and M.G.; Writing—review and editing, F.B., H.B.M. and M.G. All authors have read and agreed to the published version of the manuscript.

**Funding:** The authors would like to extend their sincere appreciation to the Deanship of Scientific Research at King Saudi University for funding this research group (RG-1437-020).

**Acknowledgments:** The authors would like to extend their sincere appreciation to the Deanship of Scientific Research at King Saudi University for funding this research group (RG-1437-020). We thank M.E.H. Ismail for his valuable comments during the work in this paper.

**Conflicts of Interest:** The authors declare no conflict of interest.

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
