# Peer review of "On the Zeros of the Big q-Bessel Functions and Applications"

_mathematics, doi:10.3390/math8020237_

Round 1

Reviewer 1 Report

This paper presents some useful tools, which are a nice contribution in the exploration of some properties of q-Bessel functions theory.

All the paper is well written, and I have just one suggestion to improve the quality of math expressions: Some brackets in math equations should be improved. “\bigleft(“ latex command should help.

Author Response

We would like to thank the reviewer for careful and thorough reading of this manuscript, which help to improve the quality of this manuscript.

We have improved the quality of math expressions. 

Regards 

Bouzeffour F, and M. Garayev

Reviewer 2 Report

The paper deals with some features of particular Bessel functions, and, in particular, with their zeors. Peculiar cases are analized also in the asymptotic regime.
Both the compilative parts and the orginal-research results are very-well presented and very well- exposed;
interesting examples complete the exposition.
Tha manuscript would be greatly improved by the following modifications, as for the style standards of the Journal, as well as
for the sake of spread and increase of bibliometrical indice
and for the profitabileness by the readers and researchers of contiguous pertinent research fileds and guidelines.

Minor correction 1) The mansucript lacks of an Outlook/Concluding Remark Section. Such a section should contain both the
summarization of the main original findings of the manuscript, as well as the example proposed and bibliographical items
which should complete some possible directions of the topics studied and possible future applications, according of
the choices of the Author.

Minor correction 2) The Introduction Section is complete and precise as far as the topics studied are concerned. The Introduction
Section does not contain much information about applications and Reference items present in the Literature, as well as the description
of possible applications.

I do encourage publication after the minor revisions.

Overall Editorial Suggestion: Minor Revisions.

Author Response

Thank you for your  opportunity to revise our paper

 We have included all comments and remaks of the reviewer  in the new version of the manuscript.

Regards 

F. Bouzeffour and M. Garaey

Reviewer 3 Report

Dear Sir,

I have checked the paper. My comments are listed below.

1-Motivations must be improved.

2-Although the mathematical analysis are well-done, we need to know the implications of all these beautiful equations in applied mathematics. The illustration is correct, however I recommend one more discussion or illustration dealing with the implications of these equations in applied sciences.

Technically all equations are correct but points 1 and 2 must be addressed.

A minor revision is required. 

Bests

Author Response

Thank you for your  opportunity to revise our paper

 We have included all comments and remaks of the reviewer  in the new version.

Regards 

F. Bouzeffour and M. Garaey